# Computational Optimization of Irradiance and Fluence for Interstitial Photodynamic Therapy Treatment of Patients with Malignant Central Airway Obstruction

**DOI:** 10.3390/cancers15092636

**Published:** 2023-05-06

**Authors:** Emily Oakley, Evgueni Parilov, Karl Beeson, Mary Potasek, Nathaniel Ivanick, Lawrence Tworek, Alan Hutson, Gal Shafirstein

**Affiliations:** 1Department of Cell Stress Biology, Photodynamic Therapy Center, Roswell Park Comprehensive Cancer Center, Buffalo, NY 14263, USA; 2Simphotek, Inc., 211 Warren St., Newark, NJ 07103, USA; 3Department of Thoracic Surgery, Roswell Park Comprehensive Cancer Center, Buffalo, NY 14263, USA; 4Department of Biostatistics and Bioinformatics, Roswell Park Comprehensive Cancer Center, Buffalo, NY 14263, USA

**Keywords:** interstitial photodynamic therapy, I-PDT, malignant central airway obstruction, MCAO, treatment planning, computational optimization, irradiance, fluence, rate-based light dose

## Abstract

**Simple Summary:**

There are no effective treatments for patients with cancers that induce airway narrowing via extrinsic pressure to the bronchus (i.e., extrinsic malignant central airway obstruction—MCAO). The effects of these cancerous tumors must be quickly alleviated to allow normal breathing and delay disease progression. Currently, stents are used to keep the airway open, but stents do not halt the progression of the cancerous tumor that can crush the stent. We have shown that interstitial photodynamic therapy (I-PDT) can be a safe and beneficial treatment option for patients with extrinsic MCAO. Image-based pre-treatment planning is critical for patient safety and tumor response in I-PDT. Herein, we present and validate novel image-based computer optimization methods for guiding light administration in I-PDT of extrinsic MCAO, based on a rate-based light dose metric. We demonstrate the benefit of our approach in data from representative patients with extrinsic MCAO who were treated with I-PDT.

**Abstract:**

There are no effective treatments for patients with extrinsic malignant central airway obstruction (MCAO). In a recent clinical study, we demonstrated that interstitial photodynamic therapy (I-PDT) is a safe and potentially effective treatment for patients with extrinsic MCAO. In previous preclinical studies, we reported that a minimum light irradiance and fluence should be maintained within a significant volume of the target tumor to obtain an effective PDT response. In this paper, we present a computational approach to personalized treatment planning of light delivery in I-PDT that simultaneously optimizes the delivered irradiance and fluence using finite element method (FEM) solvers of either Comsol Multiphysics^®^ or Dosie™ for light propagation. The FEM simulations were validated with light dosimetry measurements in a solid phantom with tissue-like optical properties. The agreement between the treatment plans generated by two FEMs was tested using typical imaging data from four patients with extrinsic MCAO treated with I-PDT. The concordance correlation coefficient (CCC) and its 95% confidence interval (95% CI) were used to test the agreement between the simulation results and measurements, and between the two FEMs treatment plans. Dosie with CCC = 0.994 (95% CI, 0.953–0.996) and Comsol with CCC = 0.999 (95% CI, 0.985–0.999) showed excellent agreement with light measurements in the phantom. The CCC analysis showed very good agreement between Comsol and Dosie treatment plans for irradiance (95% CI, CCC: 0.996–0.999) and fluence (95% CI, CCC: 0.916–0.987) in using patients’ data. In previous preclinical work, we demonstrated that effective I-PDT is associated with a computed light dose of ≥45 J/cm^2^ when the irradiance is ≥8.6 mW/cm^2^ (i.e., the effective rate-based light dose). In this paper, we show how to use Comsol and Dosie packages to optimize rate-based light dose, and we present Dosie’s newly developed domination sub-maps method to improve the planning of the delivery of the effective rate-based light dose. We conclude that image-based treatment planning using Comsol or Dosie FEM-solvers is a valid approach to guide the light dosimetry in I-PDT of patients with MCAO.

## 1. Introduction

Photodynamic therapy (PDT) is a binary treatment that involves the administration of a photosensitizer that can be activated with visible or near-infrared light to generate reactive oxygen species, typically singlet oxygen, that induces irreversible cell damage [1,2,3]. The response to PDT is governed by the photosensitizer type and concentration, light dose rate (irradiance) and dose (fluence), tumor oxygenation, and in some cases immune response [1,2,4]. In the context of treating large and deep-seated tumors, interstitial photodynamic therapy (I-PDT) must be employed to induce an effective photoreaction [5,6]. In this setting, the systemic administration of the photosensitizer is followed by intratumoral illumination administered by a laser through optical fiber/s that are inserted directly into the target tumor [5].

Pre-treatment planning for computing the light fluence in the target tumor is a critical component contributing to patient safety and response following I-PDT [5,6,7,8]. We and others have shown that image-based computer simulations of light propagation are required to estimate the intratumoral light fluence in I-PDT [6,7,8,9,10,11]. Several groups have reported on the use of computer-based simulations to plan and optimize the light delivery before and during I-PDT from bare-end fibers (as point light sources) [7,8], or cylindrical diffuser fibers (CDF) [6,12,13,14,15]. These methods assume that the fibers are arranged in a uniform pattern, usually a parallel placement of fibers into the target tumor. However, this is not always applicable for deeply seated cancers surrounded by critical structures, which require more complex fiber configurations. One paper presented automatic treatment planning, based on convex optimization, that maximizes the delivered light fluence and that is applicable to arbitrary configurations and applied to a brain tumor model [12]. However, all these methods only looked at delivering a therapeutic light fluence to the tumor volume.

In our recent preclinical studies, we used an image-based finite element method (FEM) to guide I-PDT light delivery, which demonstrated that in addition to PS tumor retention, the baseline tumor volume, and fluence, the intratumoral irradiance is a key parameter indicative of tumor response to I-PDT with porfimer sodium (Photofrin^®^, Pinnacle Biologics Inc., Bannockburn, IL, USA) [10,16]. In external beam PDT of superficial tumors, low irradiance at the tumor surface is the key parameter for effective response PDT [17,18]. For I-PDT of locally advanced cancers in mice and rabbits, we reported that exceeding the minimal irradiance at the tumor margins is the key parameter for effective response with intravenous injection of 5 mg/kg Photofrin at 24 ± 2 h prior to laser light [10,16]. Thus, we showed that irradiance of 5.0 mW/cm^2^ with a fluence of 45 J/cm^2^ at the margins will result in a significantly lower cure rate (*p* < 0.05) when compared to 8.6 mW/cm^2^ with 45 J/cm^2^ [10,16], which we define in this work as the effective rate-based light dose in Photofrin^®^-mediated I-PDT with 630 nm laser light.

Based on the preclinical data, we translated our findings into the clinic, where intravenous injection of 2 mg/kg Photofrin at 48 ± 4 h prior to laser light was used in a clinical study (NCT03735095). Our FEM image-based treatment planning was used to compute the light irradiance and fluence for Photofrin mediated I-PDT in the treatment of patients with extrinsic or mixed extrinsic/intrinsic malignant central airway obstructions (MCAO). These MCAOs are often presented between the main bronchus and the pulmonary artery, aortic arch, descending aorta, or other major blood vessels [9]. These patients are not candidates for surgery due to the advanced stage of the disease. They are also not candidates for high-dose curative radiotherapy due to the high rate of serious adverse events associated with this therapy [19,20,21,22]. We have recently reported that I-PDT of MCAO is safe with promising outcomes [9], where we applied our FEM simulations using Comsol Multiphysics^®^ (Comsol Inc., Burlington, MA, USA) to guide personalized treatment plans for light delivery.

In this paper, we provide the first detailed description of our novel FEM-based computational approach to planning the administration of the effective rate-based light dose for I-PDT in the treatment of patients with MCAO. We provide the step-by-step process for optimizing the placement of CDFs in a non-uniform configuration so that critical structures receive no more than the pre-defined safe irradiance and fluence while the tumor is illuminated with a therapeutic effective rate-based light dose. We present a newly developed domination sub-maps method in Dosie™ (Simphotek Inc. Newark, NJ, USA), a FEM package with integrated analysis and visualization tools, which optimizes the administration of safe and effective rate-based light dose in I-PDT when the spatial configuration of CDFs is already known. The clinical utilization of advancing I-PDT with our new treatment planning approach is demonstrated through post-analysis of representative data from patients treated with I-PDT in our recently reported clinical study [9].

## 2. Materials and Methods

### 2.1. FEM Light Propagation Simulations in I-PDT

The use of FEM to simulate light propagation in tumors was previously described in [16,23,24]. The governing equations for irradiance $\phi \left(r,t\right)$ (W/m^2^) are
(1)$$\frac{1}{c_{n}}\left(\frac{\partial }{\partial t}\phi \left(r,t\right)-\nabla \left(\alpha_{n}\nabla \phi \left(r,t\right)\right)\right)=-\mu_{a}\phi \left(r,t\right),$$
where
(2)$$\alpha_{n}=c_{n}\cdot {\left[3\left(\mu_{a}+\left(1-g\right)\mu_{s}\right)\right]}^{-1}.$$

α_n_ is the optical diffusion coefficient (m^2^/s) of tissue n, μ_a_ and μ_s_ are the linear absorption and scattering coefficients (1/m) of tissue, g is the anisotropy factor, c_n_ is the speed of light in tissue n, and r is a position $\left(x,y,z\right)$ in the target region, r∈Ω. The right-hand side of Equation (1) represents the rate of the light energy absorbed in a unit volume (J/m^3^/s). We simulate the laser light delivered from these CDFs as the light irradiance emitted from the outside diameter of the CDF. This light source is defined by the following Dirichlet boundary condition, $r\in \partial \Omega_{laser}$:(3)$$\phi \left(r,t\right)=P_{laser}.$$

P_laser_ is the input light irradiance (W/m^2^) per source diffuser fiber, along the diffuser surface $\partial \Omega_{laser}$. It is assumed that the initial light irradiance in the tissue results from visible or near-infrared light. This initial condition is
(4)$$\phi \left(r,0\right)=P_{bg},$$
where P_bg_ is the irradiance (W/m^2^) of the background light radiation (i.e., daylight) in the target region, r∈Ω. A Robin boundary condition is applied (the vector n(r) points outside of the tissue n, which is assumed to form the boundary of the entire region Ω), r∈∂Ω:(5)$$c_{n}\phi \left(r,t\right)+\alpha_{n}\nabla \phi \left(r,t\right)\cdot n(r)=0.$$

To date, we have used Comsol to simulate light propagation in locally advanced tumors for clinical treatment planning of I-PDT [9,10,16]. Here, we present Dosie^TM^, which is a customized FEM and Monte Carlo [25,26,27,28] software developed specifically to become a treatment planning tool for I-PDT. Dosie solves the same five equations (Equations (1)–(5)) to simulate the light distribution during I-PDT. The Dosie FEM numerical module finds an approximate solution to the irradiance $\phi \left(r,t\right)$ by solving the following weak form of the governing Equation (1) that is less strict about the smoothness of a solution and holds for any test function $\psi \left(r,t\right)\in \Psi$ (which is a set of functions continuously differentiable within the target region Ω and is trivial outside of its boundary ∂Ω):(6)$$\int_{\Omega }\frac{\partial \phi }{\partial t}\psi +\int_{\Omega }\nabla \psi \cdot (\alpha_{n}\nabla \phi )+\int_{\Omega }c_{n}\mu_{a}\phi \psi +\int_{\partial \Omega }c_{n}\phi \psi =0.$$

A variant of an implicit Euler backward solver is implemented in Dosie to solve Equation (6), where the integration in Equation (6) is performed over tetrahedral finite elements. An open-source software SALOME [29] is used to generate the tetrahedral meshes, which are then tagged and imported into Dosie.

### 2.2. Image-Based FEM for I-PDT of MCAO

A step-by-step schema in Figure 1 describes the FEM-based treatment planning of the light delivery during I-PDT of MCAO. The personalized treatment plans indicate the number, location, and light settings of the CDFs needed to deliver a prescribed light irradiance and fluence to the target tumor volume. In this study, our goal was to deliver our preclinical suggested **effective rate-based light dose** (i.e., 8.6 mW/cm^2^ and 45 J/cm^2^) to a significant part of the target tumor volume. For patients’ personalized treatment planning, high-resolution diagnostic CT or MRI scans were acquired 1–2 weeks prior to I-PDT. These scans were imported into an image visualization and analysis software package (Simpleware, Exeter, UK) and used to manually segment the tumor, adjacent normal tissues, blood vessels, and other important anatomical features defined by the treating physician. The segmented scans were then reconstructed to create a 3D computer-aided design (CAD) model that was imported into the Comsol FEM package. In Comsol, cylindrical representations of the CDFs were virtually placed within the tumor volume. A tetrahedral mesh of the entire geometry, which included the tumor, the CDFs, surrounding normal tissue, and critical structures, was created in Comsol using its mesh generator with our previously determined optimal mesh parameters defined in [23].

The light propagation throughout the tumor, surrounding tissue, and the major blood vessels was computed by solving Equation (1) with the appropriate initial and boundary conditions, as shown in Equations (3)–(5). For all FEM simulations presented in this publication, we assume that the optical properties do not change during illumination. The optical properties used for the FEM simulations are used to activate the PS, Photofrin^®^, with 630 nm light. These properties are given in Table 1 and were obtained from the literature [30,31]. For each patient, a manual optimization was performed in Comsol before treatment begins (as described in the next section) to determine the number, location, and light intensity (mW/cm) of the CDFs needed to deliver the prescribed light irradiance and fluence. The resulting treatment plan was presented for approval by the treating physician.

### 2.3. Comsol Optimization of I-PDT Light Delivery in the Treatment of MCAO

In the I-PDT of MCAO, the light is delivered by inserting one CDF at a time into the tumor through the endobronchial ultrasound (EBUS) with a transbronchial needle, as described in detail in Ivanick et al., 2022 [9]. The treatment plan includes the number of fiber placements within the tumor and prescribes the milliwatt per cm (mW/cm, light intensity) and joule per cm (J/cm, light energy) for the diffuser length of each CDF. The tumor is illuminated at an intensity and energy designed to administer the therapeutic irradiance and fluence. In planning the CDFs placements, several conditions are taken into consideration. During treatment, fibers can be inserted from the bronchus and into the tumor at a 30–40° angle and no more than 4 cm deep. Additionally, the distance from the fiber to any major blood vessel (i.e., critical structure) is limited to ≥6 mm for safety, which is based on our clinical experience [9]. We aim to limit the light irradiance and fluence delivered to the blood vessels for safety reasons. Although there is no drug in the blood vessels at the time of treatment, there is a low risk of microscopic invasion of cancer cells in the vessel wall. Using these conditions, a manual optimization is performed in Comsol to determine the number and location of the CDFs. For each CDF placement, a parametric study is conducted in which the light intensity emitted from the CDF fiber varies between 80 and 400 mW/cm in increments of 20 mW/cm, and the resulting irradiance distribution throughout the tumor, as well as the major blood vessels, is computed using the FEM solution to Equation (1). The starting minimum light intensity (80 mW/cm) is based on clinical experience, while the maximum light intensity (400 mW/cm) is limited by the maximum laser light power that can be safely delivered through these clinical CDFs. If the irradiance and/or fluence delivered to major blood vessels exceeds 8.6 mW/cm^2^ or 9.5 J/cm^2^, which we defined as the maximum allowable exposures based on our clinical experience [9], the CDF is virtually repositioned within the tumor geometry and the simulation is rerun until a location is determined to be safe (i.e., the irradiance and fluence delivered to the major blood vessels do not exceed the set thresholds). Once safe light dosimetry and location are defined for each CDF, the irradiance and fluence resulting from consecutive light illuminations are computed within the entire tumor and surrounding critical structures. Our method for simulating consecutive light uses a combination of Comsol and Matlab^®^ (Matlab^®^, 2012a, MathWorks Inc., Natick, MA, USA) and was previously described [16]. This manual optimization of the CDF location and light dosimetry can take up to 1–3 days, depending on the complexity of the tumor geometry.

### 2.4. Dosie Optimization of Power Outputs and Treatment Times with a Novel Domination Sub-Maps Method

Dosie has been specifically designed to help with the process of manually optimizing the CDFs output intensities and treatment times and provide a framework for computational analysis and visualization. Using predetermined CDF coordinate locations within the tumor geometry, Dosie can be applied to optimize the power output (or, equivalently, the intensity at each point of a CDF) and the treatment time for each CDF by using a new method we term domination sub-maps. This optimization is performed for each CDF by maximizing its power output in a sub-region where the resulting irradiance from such CDF “dominates” the irradiances generated by other CDFs. The steps of this method are outlined in Figure 2. First, simulations are run in Dosie, where laser light is emitted from only one CDF at a time and the resulting irradiance distribution map $\phi_{CDF}\left(r\right)$ throughout the tumor geometry and surrounding major blood vessels (≥4 mm in diameter) is computed for each CDF. For these simulations, the power outputs from the CDFs are initially set to the same value. The resulting irradiance maps are then combined to determine the domination sub-maps for each CDF. The CDF domination sub-map is defined over a sub-region within the 3D tumor geometry and surrounding critical structures, where the light irradiance delivered from that CDF is at its maximum when compared to the other CDF irradiance distributions. For example, if there are four CDFs (A, B, C, and D), the domination sub-map of map $\phi_{A}\left(r\right)$ with regard to maps $\phi_{B}\left(r\right)$, $\phi_{C}\left(r\right)$, and $\phi_{D}\left(r\right)$ is defined as follows (per each position $r=\left(x,y,z\right)\in \Omega_{T}\cup \Omega_{CS}$ within the tumor, $\Omega_{T}$, or critical structure, $\Omega_{CS}$, geometries):(7)$${domn}_{A|B,C,D}(r)=\left\{\begin{array}{c} \phi_{A}\left(r\right),\mathrm{if}\phi_{A}\left(r\right)>\max\{\phi_{B}\left(r\right),\phi_{C}\left(r\right),\phi_{D}\left(r\right)\}\\ 0,\mathrm{otherwise} \end{array}\right..$$

Figure 3a–d shows representative domination sub-maps (for the tumor only; the surrounding critical structures are not shown) for one patient with MCAO for all CDFs, and Figure 4a shows a domination sub-map in the tumor and surrounding pulmonary artery, aortic arch, and blood vessels for one of the CDFs. Once the domination sub-maps for each CDF have been determined, the maximum irradiances throughout the critical structures within each domination sub-map are checked, such that the irradiance delivered to those structures from different CDFs will not exceed the effective irradiance of 8.6 mW/cm^2^.

For each CDF, the Threshold Under-Dose (TUD) factor is determined as the ratio between the irradiance threshold value of 8.6 mW/cm^2^ and the maximum irradiance within its domination sub-map (i.e., ${max\{domn}_{A|B,C,D}(r)\}$ for all $r\in \Omega_{CS}$ as for CDF A in the example above; also shown in Figure 4b with CDF location), calculated at critical structures:(8)$$TUD\;{Factor}_{A}=\frac{8.6}{{max\{domn}_{A|B,C,D}\left(r\right):r\in \Omega_{CS}\}}.$$

These TUD factors represent the irradiance-safe scale factors that should be applied to the initial input light intensities (or powers, in effect) of the CDFs, so that the resulting irradiance does not exceed 8.6 mW/cm^2^ at the critical structures. Thus, greater than 1 TUD factors allow the initial intensities to be increased safely without exposing the critical structures to irradiances >8.6 mW/cm^2^. When the TUD factor for a CDF is less than 1, applying this factor will reduce the initial light intensity delivered from the CDF to a value that will result in an irradiance map that is ≤8.6 mW/cm^2^ within the critical structures. The initial light intensities and irradiance distribution maps for each CDF are updated (in Dosie) based on the TUD scale factors to validate that the irradiance delivered to the critical structures is under 8.6 mW/cm^2^. Irradiance is linearly proportional to CDF power, so that there is no need to re-run time-consuming FEM calculations when the irradiance maps need to be updated.

The next step is to investigate the fluence (J/cm^2^) that is delivered to the critical structures using the updated light intensities for each CDF. The goal is to limit the total fluence delivered to the critical structures (i.e., major blood vessels) to ≤9.5 J/cm^2^. To achieve this, the total light fluence, $F\left(r\right)=\sum_{CDF}\phi_{CDF}\left(r\right)\cdot T_{CDF}$, delivered to all the critical structures resulting from the updated CDF light intensities is computed assuming the initial treatment time, $T_{CDF}$, per CDF. If the total fluence to any of the critical structures exceeds 9.5 J/cm^2^, the CDF domination sub-maps for the light fluence, $F_{CDF}\left(r\right)=\phi_{CDF}\left(r\right)\cdot T_{CDF}$, need to be evaluated to identify the CDF(s) that is/are contributing to the high fluence at the critical structures. The light intensity emitted from these CDFs are scaled down appropriately until the resulting total fluence delivered to the critical structures is ≤9.5 J/cm^2^. At this point, the CDF intensities, $I_{CDFs}^{*}$, have been identified to be irradiance safe and fluence safe at the critical structures.

In addition to the limitations imposed on the critical structures, there are also limitations regarding the output power delivered from the CDFs. As previously stated, the maximum light intensity that can be emitted from the CDFs is 400 mW/cm. Additionally, using the clinical laser system, the output power emitted from the CDFs can only be set and calibrated in increments of 20 mW. Based on these criteria and starting with the CDF critical-structure-safe intensities $I_{CDFs}^{*}$ identified in the previous steps, the output power from each CDF is calculated and safely increased to its maximum in the steps of 20 mW (if the powers are in between the 20 mW increment, the power is scaled down to the closest power that is a factor of 20 mW). Critical-structure-safe fluence values, along the CDFs surfaces, are also calculated from $I_{CDFs}^{*}$. Dividing the resulting Critical-structure-safe powers by the Critical-structure-safe fluences and multiplying by the CDF areas give us the critical-structure-safe treatment times.

The initial light intensities and irradiance distribution maps for each CDF (using the Dosie expressions calculator) are updated based on the optimized critical-structure-safe CDF powers and treatment times to validate that the maximum irradiance delivered to the critical structures is ≤8.6 mW/cm^2^ and that the total fluence accumulated at the critical structures from all CDFs is ≤9.5 J/cm^2^. The previous steps of identifying safe powers and treatment times for CDFs can be repeated as many times as needed, as long as the resulting irradiance and the total fluences at the critical structures are below the thresholds of 8.6 mW/cm^2^ and 9.5 J/cm^2^, respectively.

The resulting optimized powers and treatment times for CDFs are used to update the initial irradiance distribution maps, which will be used to compute the irradiance and the total effective fluence DVHs for the target tumor volume. In this method, Dosie FEM calculations used to estimate the overall irradiance map $\phi \left(r\right)$ are performed only once per each CDF, while all the updates to the irradiance are completed with the Dosie expressions calculator based on the new power and treatment time assignment.

## 3. Results

### 3.1. Validation of the FEM Solvers in an Optical Phantom

We validated Comsol and Dosie by comparing each of the simulation results with light dosimetry measurements in a polyurethane solid phantom with known optical properties (Biomimic^TM^ Optical Phantoms, INO, Québec, Canada). The phantom linear absorption coefficient at 630 nm, μ_a,_ was 0.224 (1/cm); the phantom reduced scattering coefficient at 630 nm, μ_s_′, was 4.99 (1/cm). The light dosimetry measurements were conducted with our calibrated light dosimetry system including the necessary correction factors for differences in the index of refractions as previously described in Oakley et al. 2015 [23]. The irradiance was measured with the system’s isotropic detection fibers (IP85, Medlight SA, Ecublens, Switzerland) which were inserted into the solid phantom at 5, 10, 15, 20, and 25 mm away from a 2 cm CDF emitting 630 nm light at 100 mW/cm (total 200 mW).

The agreement between the phantom measurements and the Comsol simulations and between the phantom measurements and the Dosie simulations was determined by calculating the concordance correlation coefficients (CCC) [32]. Values of the CCC are within the range of −1 to 1, with 1 denoting perfect agreement and 0 denoting a lack of agreement. Table 2 shows the results of this phantom experiment. The CCC between the measurements and the Comsol simulations was 0.999 (95% CI, 0.985–0.999). The CCC between the measurements and the Dosie simulations was 0.994 (95% CI, 0.953–0.996). These results demonstrate excellent agreement between the simulations and measurements.

### 3.2. Treatment Planning for I-PDT of MCAO Using Comsol

A representative example of the Comsol FEM-based treatment planning for light delivery during I-PDT for a patient with MCAO is shown in Figure 5. A CT scan of the patient was used to segment out the tumor, pulmonary artery, aortic arch, bronchus, vertebra, and the surrounding normal tissue. For this patient, the tumor volume was 26.1 cm^3^. Based on the FEM simulations, four CDFs with an illumination length of 1.5 cm and each delivering 240 mW/cm were required in order to deliver ≥8.6 mW/cm^2^ to 99.97% of the tumor volume. Figure 6 shows the resulting light irradiance distribution for each CDF placement that was determined using the Comsol optimization. In the 3D irradiance distributions for the tumor volume, anything in red received ≥8.6 mW/cm^2^. For this patient, the treatment time was set to 750 s per fiber placement. Based on the four fiber placements, the patient would have received the effective rate-based light dose to 79.6% of the tumor volume. The maximum irradiance and fluence delivered to the pulmonary artery were 7.6 mW/cm^2^ and 6.0 J/cm^2^. The maximum irradiance and fluence delivered to the aortic arch were 0.6 mW/cm^2^ and 0.5 J/cm^2^.

### 3.3. Dosie Is in Agreement with Comsol FEM Simulations for I-PDT of MCAO

Using the same tumor geometry, CDF positions, and light intensity and energy as the initial treatment plans prepared for four patients with MCAO, FEM software was used to compute the resulting irradiance and fluence dose volume histograms (DVHs) for the target tumor volume and surrounding critical structures (i.e., major blood vessels). The CCC with 95% CI (as described in Section 3.1) was used to compare the Dosie simulated DVHs to the Comsol simulated DVHs.

Of the four patients, one patient had five different tumor locations while the other three patients had one tumor location. Tumor volumes ranged from 0.7–26.1 cm^3^. Figure 7 shows the resulting rate-based light dose DVH for the tumor geometries that were computed from the Comsol and Dosie FEM simulations. The irradiance dose rate volume histogram (DRVH) is defined and computed as the percent tumor volume that received ≥8.6 mW/cm^2^. The effective rate-based light dose DVH is defined and computed as the percent tumor volume that received ≥8.6 mW/cm^2^ and ≥45 J/cm^2^. There was near perfect agreement between the two FEM software when calculating the intratumoral irradiance DRVH with a CCC of 0.999, 95% CI (0.996, 0.999). There was very good agreement between the two FEM software when calculating the effective rate-based intratumoral light dose DVH with a CCC of 0.977, 95% CI (0.916, 0.987).

### 3.4. Dosie Has the Potential to Improve the Administration of Rate-Based Light Dose

We generated another set of treatment plans for the four patients using Dosie’s domination sub-maps method to optimize power allocation and treatment time for each CDF. The CDF positions for the patients were taken from the original treatment planning conducted using Comsol FEM simulations. We applied Dosie’s domination sub-maps method to demonstrate whether or not we could adjust the CDF output power and treatment time in order to increase the intratumoral irradiance and effective fluence distribution. The results are shown in Table 3. The Dosie optimization method could increase the total effective rate-based light dose DVH by 9.8%, 6.5%, 3.1%, 15%, 9.6%, and 10.7% for, respectively, P1/T1, P1/T2, P1/T3, P1/T5, P3, and P4.

## 4. Discussions

In this paper, we described the manual optimization method employed in Comsol to determine the fiber position and power settings for I-PDT of MCAO. In addition, we introduced another FEM software, Dosie (Simphotek, Inc., NJ). The importance of Dosie is that it was designed specifically to become a medical computational device for treatment planning of light delivery during I-PDT. In this paper, we demonstrated how to use Dosie to run power allocation optimization for a known CDFs configuration using Dosie’s Domination sub-maps method, as introduced in this paper. We validated the two software packages in a solid phantom with calibrated optical properties, where the simulated light distributions from the FEMs were in agreement with dosimetry measurements taken in the solid phantom (CCC = 0.999 for Comsol and CCC = 0.994 for Dosie). Then, we used four 3D models obtained from patients with MCAO to compare the two software packages. Using the same fiber positions and light settings, we showed that Dosie is in agreement with Comsol when calculating the irradiance and the rate-based light dose distribution throughout the tumor geometries (CCC = 0.999 for irradiance calculation and CCC = 0.977 for a rate-based light dose). We conclude that Dosie can be used to accurately simulate the light irradiance and fluence during I-PDT of MCAO. Any difference in the two FEM simulations can be attributed to the difference in the 3D tetrahedral meshes used. Comsol has its own built-in meshing toolbox while Dosie uses open-source software, SALOME, to generate and annotate the tetrahedral meshes.

In our current clinical trial for the I-PDT of MCAO, we have been able to develop treatment plans using our image-based FEM in Comsol that would deliver the effective irradiance (≥8.6 mW/cm^2^) to 70–100% of the target tumor volume. However, due to the location of the major critical structures and restrictions on treatment time, we have not always been able to develop a plan that would deliver the effective rate-based light dose to the majority of the target tumor volume. Applying Dosie’s Domination sub maps method to optimize CDF power output and treatment time, we may be able to increase the percentage of the tumor volume that receives the effective rate-based light dose by as much as 15%, thereby possibly improving the treatment outcome. Future studies will investigate the use of Dosie’s domination sub-maps method in a clinical setting for power allocation and treatment time optimization, as well as fiber placement.

## 5. Conclusions

Image-based FEM can be used to guide the delivery of a therapeutic light irradiance and fluence for I-PDT in the treatment of patients with MCAO. Comsol Multiphysics^®^ and/or Dosie™ can be used to strategize a treatment plan for administering our recommended effective rate-based light dose (i.e., the irradiance and fluence of, respectively, ≥8.6 mW/cm^2^ and ≥45 J/cm^2^) while keeping the irradiance and the fluence at the surrounding critical structures (e.g., major blood vessels) below safety thresholds. We presented detailed description of treatment planning procedure using either Comsol or Dosie for MCAO I-PDT. Dosie’s newly developed domination sub-maps method can improve the planning of the delivery of the effective rate-based light dose. The presented optimization methodology is fairly general and can be potentially adopted for other indications and photosensitizers in I-PDT treatment planning, assuming that the appropriate rate-based light doses are provided.

## Figures and Tables

**Figure 1 cancers-15-02636-f001:**
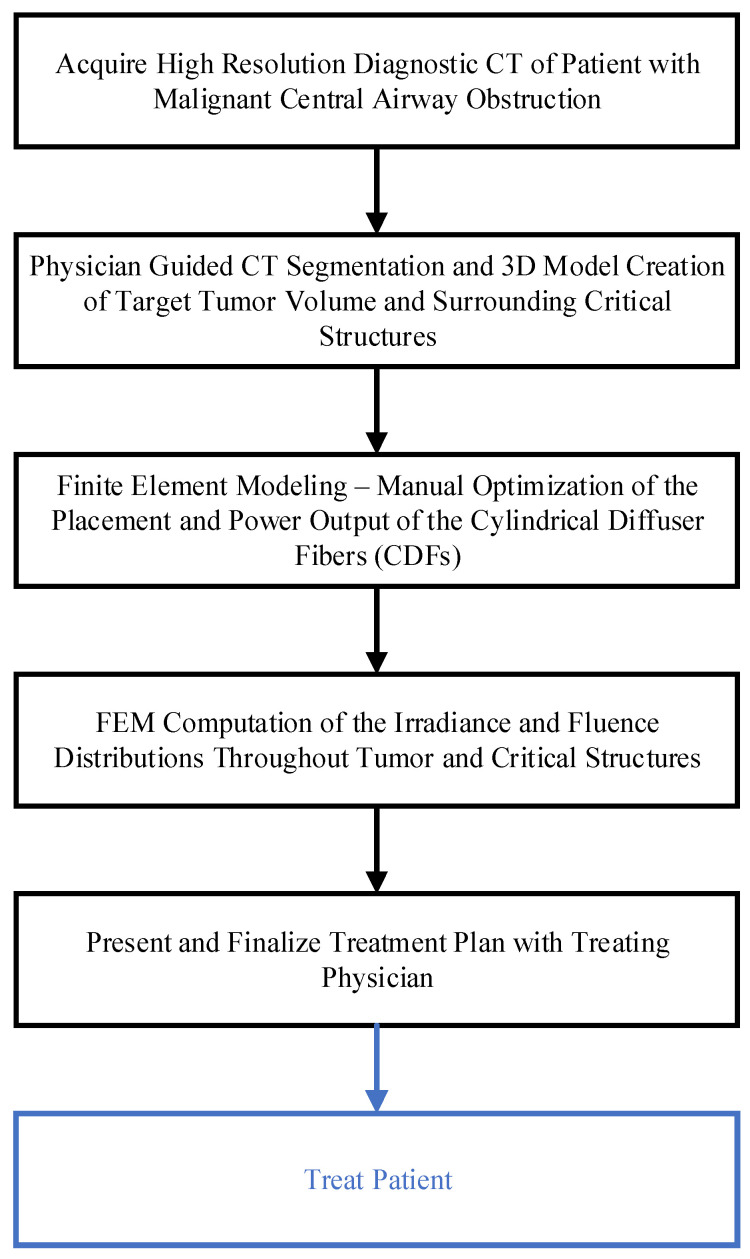
Schema of the step-by-step treatment planning procedure for I-PDT.

**Figure 2 cancers-15-02636-f002:**
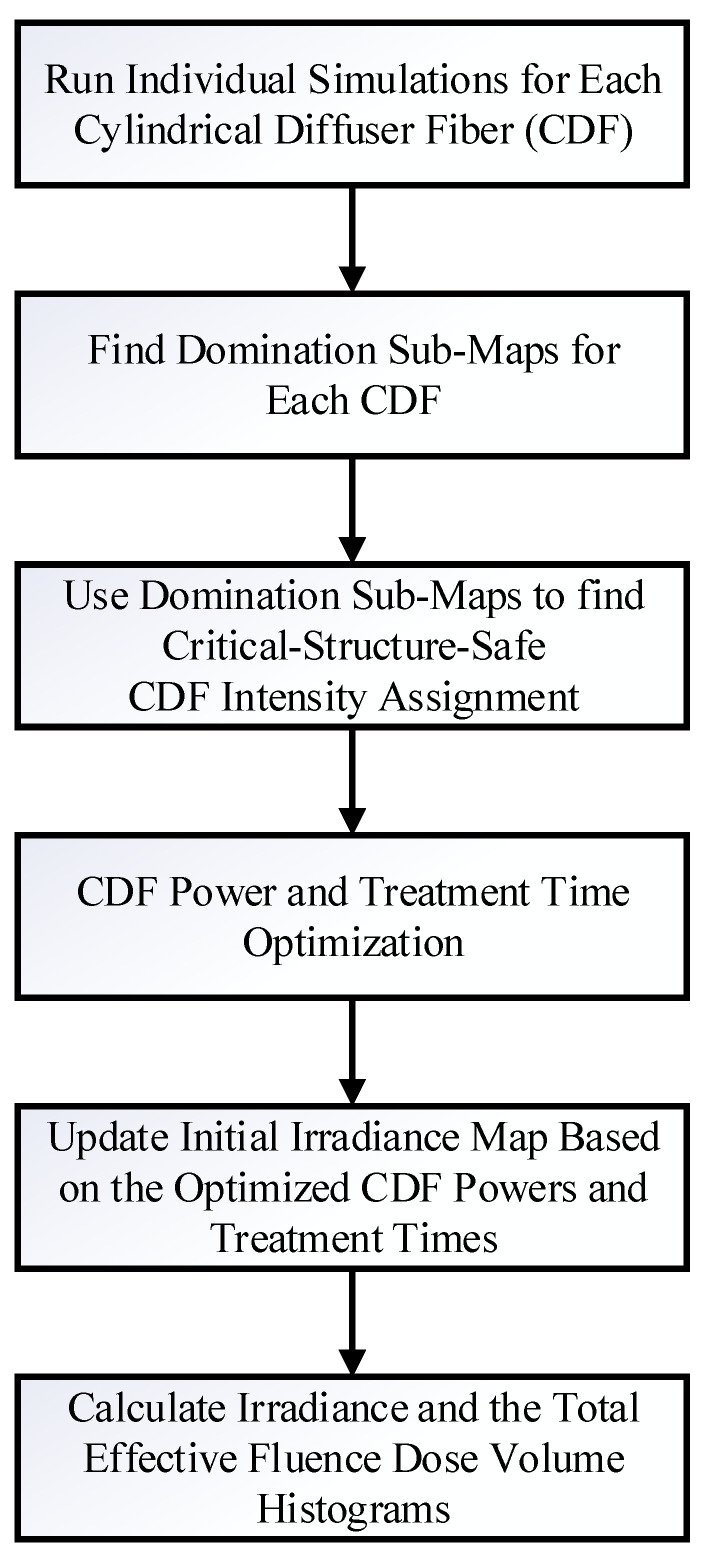
Dosie Power Optimization based on Domination Sub-maps method.

**Figure 3 cancers-15-02636-f003:**
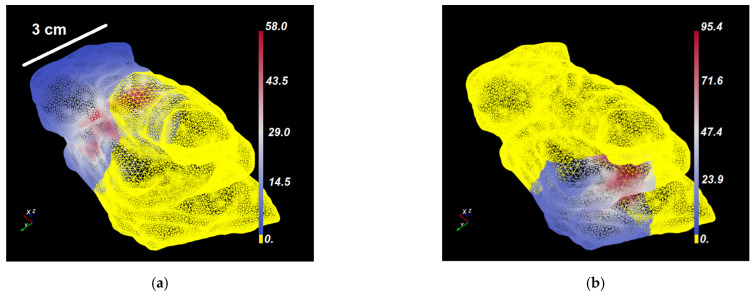
(**a**–**d**) Dosie screenshots of Domination Sup-maps (i.e., irradiance in mW/cm^2^) over the tumor of a typical patient with locally advanced MCAO tumor for four CDFs: (**a**) CDF-A; (**b**) CDF-B; (**c**) CDF-C; and (**d**) CDF-D. Non-yellow colors in (**a**–**d**) represent regions where a particular CDF dominates the other three CDFs. Power allocation optimization is carried within each such domination sub-map and then the results are combined to see if the effective rate-based light dose is reached in the presence of all CDFs. (**e**) A fragment of the patient’s 3D mesh with tumor and surrounding critical structures that shows locations where CDFs A–D were inserted.

**Figure 4 cancers-15-02636-f004:**
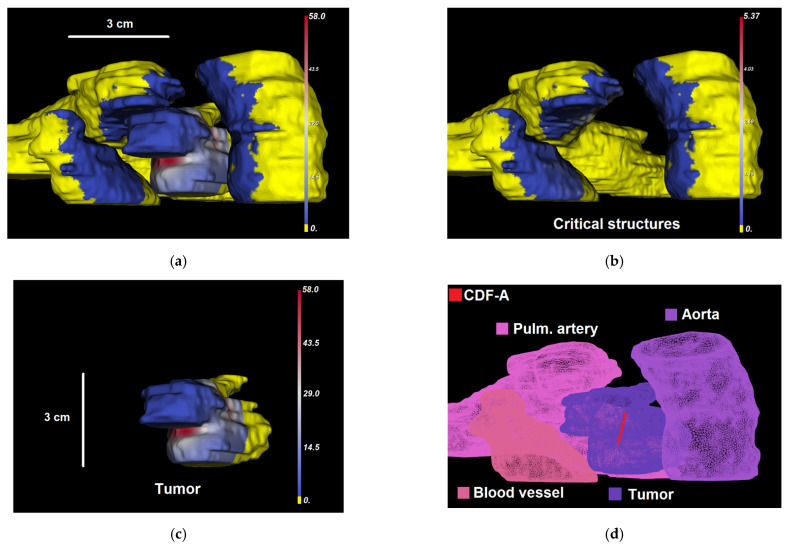
(**a**–**c**) Dosie selective screenshots of Domination Sup-map for CDF A, ${domn}_{A|B,C,D}(r)$, over (**a**) the tumor and critical structures ($r\in \Omega_{T}\cup \Omega_{CS}$); (**b**) critical structures only ($r\in \Omega_{CS}$); and (**c**) the tumor only ($r\in \Omega_{T}$). In a–c, regions that are colored yellow are dominated by CDF A. Other regions show the corresponding calculated domination sub-maps values in mW/cm^2^. The snapshots are taken from a different viewing angle than in Figure 3. (**d**) Shows labeled organs’ surfaces and CDF-A location within 3D mesh.

**Figure 5 cancers-15-02636-f005:**
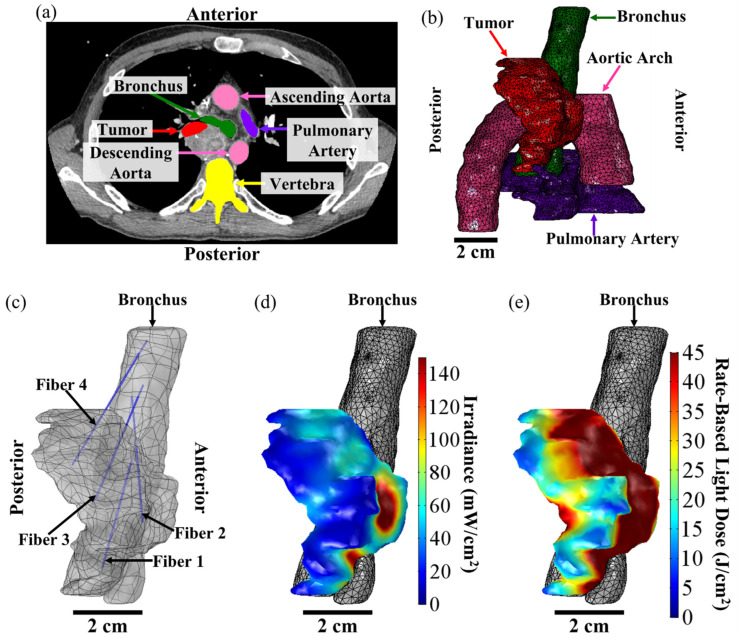
FEM-Based Treatment Plan for I-PDT of MCAO. The above figure shows a representative treatment plan and simulation results for a patient with MCAO that was treated with I-PDT. (**a**) High-resolution diagnostic CT scan. The treating physician segmented the tumor treatment volume (in red) and then the surrounding critical structures were segmented. The total tumor volume was 26.1 cm^3^. For this patient, segmentations were performed for the pulmonary artery (in purple), ascending and descending aorta (in pink), bronchus (in green), and vertebra (in yellow), and then for the tissue surrounding the tumor (not shown). These segmentations were used to create 3D CAD models that could be imported into the FEM software, which in this case was Comsol. (**b**) Three-dimensional mesh of the tumor (in red), pulmonary artery (in purple), aortic arch (in pink), and bronchus (in green). The mesh was created from the 3D CAD models in Comsol and used for our FEM simulations of light distribution. For this patient, the light irradiance and fluence distribution was computed for the tumor, surrounding normal tissue, pulmonary artery, and the aortic arch. (**c**) Fiber placement from the bronchus and into the tumor geometry. The plan was to insert 4 CDFs with illumination lengths of 1.5 cm. Each fiber would emit 240 mW/cm for 750 s. (**d**) Resulting irradiance distribution. Based on the treatment plan, 99.97% of the tumor volume would receive ≥8.6 mW/cm^2^. The irradiance ranged 7–1528 mW/cm^2^. (**e**) Resulting rate-based light dose (the total fluence calculated when the irradiance ≥ 8.6 mW/cm^2^). Based on the treatment plan, 79.6% of the tumor volume would have received the effective rate-based light dose (i.e., ≥8.6 mW/cm^2^ and 45 J/cm^2^). This volume of the tumor is indicated in red in (**e**).

**Figure 6 cancers-15-02636-f006:**
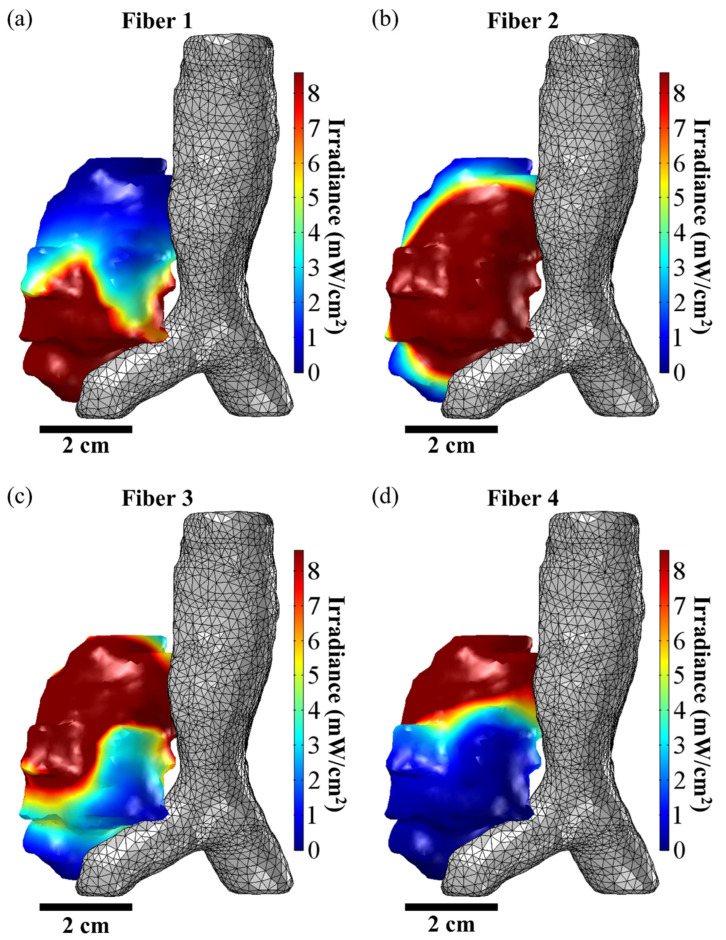
FEM-Based Manual Optimization of Fiber Placement. The above figure shows the irradiance distribution from the placement of the four CDFs used to treat a representative patient with MCAO who was treated with I-PDT. (**a**–**d**) show the resulting irradiance distribution for, respectively, fiber positions 1, 2, 3, and 4 when the input light intensity was 240 mW/cm per fiber. The goal of the treatment planning was to deliver ≥8.6 mW/cm^2^ to the tumor volume. In the figures, the volume of the tumor that is ≥8.6 mW/cm^2^ is given in red. Based on the simulations, 46.4%, 57.6%, 69.6%, and 48.3% of the tumor volume will receive ≥8.6 mW/cm^2^ when light is emitted from, respectively, Fiber 1, Fiber 2, Fiber 3, and Fiber 4.

**Figure 7 cancers-15-02636-f007:**
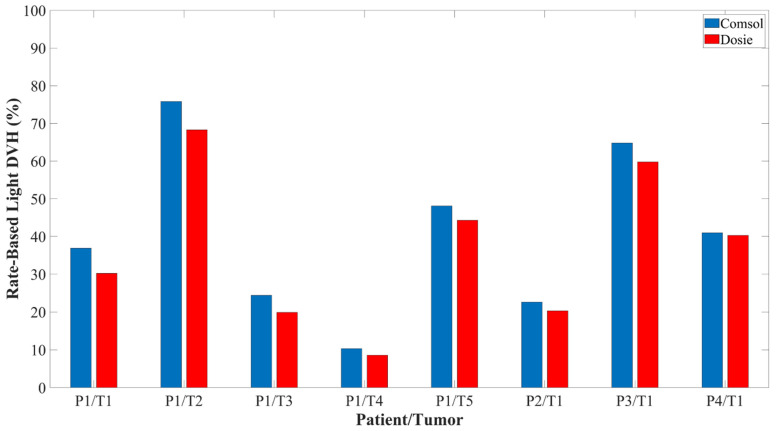
The effective rate-based light dose volume histogram calculated for 4 representative patients using both FEM software packages.

**Table 1 cancers-15-02636-t001:** The FEM simulation used tumor optical properties for 630 nm light taken from Robinson et al., 2010 [30]; the blood optical properties were taken from Van Germert et al., 1995 [31].

Title 1	Input Data	Description
Tumor Tissue and Surrounding Normal Tissue	μ_a_ = 0.2 (1/cm) μ_s_′ = 5.0 (1/cm) n = 1.37	Tissue linear absorption coefficient Tissue reduced scattering coefficient Tissue refractive index
Pulmonary Artery, Descending Aorta, and Other Major Blood Vessels	μ_a_ = 50.0 (1/cm) μ_s_′ = 2.32 (1/cm) n = 1.33	Blood linear absorption coefficient Blood reduced scattering coefficient Blood refractive index

**Table 2 cancers-15-02636-t002:** Validating FEM Solvers in Phantom Experiment.

Distance from CDF (mm)	Average Measured Irradiance (mW/cm^2^)	Comsol Computed Irradiance (mW/cm^2^)	Dosie Computed Irradiance (mW/cm^2^)
5	46.7 ± 3.4	47.0	50.5
10	15.1 ± 0.8	13.6	12.7
15	4.7 ± 0.2	4.4	4.0
20	1.4 ± 0.09	1.5	1.3
25	0.4 ± 0.03	0.5	0.5

**Table 3 cancers-15-02636-t003:** Dosie Domination Sub-Maps Method for Optimizing CDF Power and Treatment Time.

Patient	CDF Power Output Comsol → Dosie	Time Comsol → Dosie	Irradiance DRVH Comsol → Dosie	Effective Rate-Based Light Dose DVH Comsol → Dosie
P1/T1	120 mW → 200 mW	500 s → 520 s	99.5% → 100%	36.9% → 46.7%
P1/T2	100 mW → 120 mW	500 s → 500 s	100% → 100%	75.8% → 82.3%
P1/T3	120 mW → 100 mW	500 s → 560 s	91.2% → 90.5%	24.4% → 27.5%
P1/T4	120 mW → 120 mW 120 mW → 160 mW	500 s → 520 s 500 s → 540 s	38.1% → 40.5%	10.3% → 10.3%
P1/T5	80 mW → 100 mW	500 s → 500 s	100% → 100%	48.1% → 63.1%
P2	120 mW → 100 mW	750 s → 900 s	72.7% → 69.3%	22.6% → 20.3%
P3	300 mW → 400 mW 300 mW → 600 mW 300 mW → 600 mW 300 mW → 160 mW	750 s → 740 s 750 s → 900 s 750 s → 840 s 750 s → 760 s	92.8% → 93.4%	64.8% → 74.4%
P4	120 mW → 100 mW 120 mW → 100 mW	750 s → 720 s 750 s → 900 s	70.4% → 76.0%	41.0% → 51.7%

## Data Availability

The following data will be shared: participants data that motivate the results reported in this article, after deidentification (text, tables, and figures). The data will be available beginning at 9 months and ending 36 months after the publication of the article. The data will be provided to researchers who provide a methodologically sound proposal to achieve the aims of the approved proposals. The proposals should be directed to Nathaniel Ivanick MD at nathaniel.ivanick@roswellpark.org and Gal Shafirstein, D.Sc. at gal.shafirstein@roswellpark.org. The data will be provided through agreement with Roswell Park Comprehensive Cancer Center.

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
