# Peer review of "Computational Optimization of Irradiance and Fluence for Interstitial Photodynamic Therapy Treatment of Patients with Malignant Central Airway Obstruction"

_cancers, 2023, doi:10.3390/cancers15092636_

Round 1
Reviewer 1 Report
The authors report a new software package for the 3D calculation of intensity and dose of an interstitial PDT treatment. It is, of course, a matter of the subject that such program packages are are used as a black box and are judge only by the outcome. At least thats the transported message. If it was the intention of the authors to tell more about the working principle, the materials/methods part needs an improvement beyond the general mathematical description of the light propagation in diffuse media. The article is written in a very complicated style. The english is likely very much better than mine, still it took me several attempts to understand the message. It seems to go in circles. What I understood is that the new programm is superior to the commonly known one, because of the ability to calculate a variable linear combination of the included sub masks. I was quite surprised to hear that this is regarded a special feature. Because of this new feature, a better illumination of the tumor can be reached, which means a higher volume percentage is illuminated sufficiently íntensive and with enough dose. If I got it right, why did it take 15 pages to give this information?
Yes, the program packages are black boxes, your input values should not be the same.
Which wavelentgh is used? For the phantom calculations, 630 nm are mentioned, but is that the wavelength used for treatments? It is quite annoying to read 8.6 again and again. Please explain the input parameters at the beginning - and give reasons - and then use short illustrative phrases.
The 8.6 mW/cm² might be correct for a certain contentration (which?) of Photofrin (?) at 630 nm (?) with an extinction of (?) and a singlet oxygen quantum yield of (?), resulting in how many mol singlet oxygen /cm³s (?).
Without giving all this information the 8.6 mW/cm² is worth nothing.
Just change one parameter and the min. illumination rate will be a different one. But repating it a hundred times, makes it sound like a magic number.
A side condition of the plan is to avoid strong illumination of things like vessels. Again, the reason is a black box: experience. But there are reasons for that. In blood there is so much more oxygen available and Photofrin might be handy but from the stand point of targeted delivery it is horrible. Consequently, lots of drug is circulating in the blood, which consequently causes strong damadges under illumination...more then anywhere else.
I want to see this draft published, but please sort it. Make it easy to read and make it sound less like an advertisment.
Further comments:
Please, describe the setup once only.
Can you give a lenght scale in images 3 and 4 too, please?
The irradiance ranged 7.0 – 1527.9 mW/cm2. May be the simulation tells you 5 digits, but reduce them.
The very high intensities are probably in contact to the fibers, but 1.5W/cm² appears a bit much. Aren't there upper limits for efficient treatment, aswell?
Your references are very american.
Author Response
Please see the attachment.
Please, be aware that the Lines that are provided in the response (Lines 273-274, 86-96, 294-297) as the reference to the places where the changes have been made, may not be the same when opened in different versions of MS Word and different platforms (Win/Mac). That is why, for each such reference, we also provided the fragment from the paper where the correspondent change begins.

Reviewer 2 Report
Computational Optimization of Irradiance and Fluence for Interstitial Photodynamic Therapy Treatment of Patients with Malignant Central Airway Obstruction was proposed by Emily Oakley et al. and their coworkers. Their explanations and mathematical simulation techniques have opened an avenue in the treatment of PDT cancer with minimal toxicity and a high rate of tumor destruction. I personally appreciate their clinical studies, methods, results, and the way they describe potential benefits.
I would recommend that this effort be published in its current form.
Author Response
There was nothing to address here.
Round 2
Reviewer 1 Report
Thanks to the authors for the explanations and the changes made. I still find the value repetition annoying, but OK I can live with it.